# Human Allogeneic Liver-Derived Progenitor Cells Significantly Improve NAFLD Activity Score and Fibrosis in Late-Stage NASH Animal Model

**DOI:** 10.3390/cells11182854

**Published:** 2022-09-13

**Authors:** Mustapha Najimi, Sébastien Michel, Maria M. Binda, Kris Gellynck, Nathalie Belmonte, Giuseppe Mazza, Noelia Gordillo, Yelena Vainilovich, Etienne Sokal

**Affiliations:** 1Cellaïon, 1435 Mont-Saint-Guibert, Belgium; 2UCLouvain, Laboratory of Pediatric Hepatology and Cell Therapy (PEDI), Institute of Experimental and Clinical Research (IREC), 1200 Brussels, Belgium

**Keywords:** NASH, HALPCs, liver, paracrine effects, STAM mouse, cell therapy, immunomodulation

## Abstract

Accumulated experimental and clinical evidence supports the development of human allogeneic liver-derived progenitor cells (HALPCs) to treat fibro-inflammatory liver diseases. The aim of the present study was to evaluate their therapeutic effect in a non-alcoholic steatohepatitis (NASH)-STAM mouse model. The immune signaling characteristics of HALPCs were first assessed in vitro. Upon inflammation treatment, HALPCs secreted large amounts of potent bioactive prostaglandin E2 and indoleamine 2,3-dioxygenase, which significantly reduced CD4^+^ T-lymphocyte proliferation and secretion of proinflammatory cytokines. In vivo, HALPCs were intravenously administered as single or triple shots (of a dose of 12.5 × 10^6^ cells/kg BW) in STAM mice. Transplantation of HALPCs was associated with a significant decrease in the NAFLD activity score at an early stage and in both inflammation and hepatocyte ballooning scores in late-stage NASH. Sirius red staining analyses revealed decreased collagen deposition in the pericentral region at both stages of NASH. Altogether, these findings showed the anti-inflammatory and anti-fibrotic features of HALPCs in an in vivo NASH model, which suggests their potential to reverse the progression of this chronic fibro-inflammatory disease.

## 1. Introduction

Non-alcoholic fatty liver disease (NAFLD) associated with excessive lipid accumulation in the liver is becoming the leading cause of chronic liver disease. It is a major health issue due to its close association with the worldwide epidemics of obesity and diabetes. NAFLD covers a spectrum of hepatic lesions in the absence of alcohol intake, ranging from pure steatosis (non-alcoholic fatty liver) to steatosis with hepatocellular injury and inflammation (non-alcoholic steatohepatitis or NASH). It initially develops as a consequence of significant accumulation of triglycerides in hepatocytes, followed by oxidative stress and proinflammatory cytokine activation that ultimately evolves to fibrosis [1,2,3]. If not stopped or treated, fibrosis may progress towards irreversible end stages of liver failure, cirrhosis, and cancer. NAFLD is also a risk factor for the development of renal, cardiovascular, and cerebrovascular disorders [4]. Accordingly, early management of NAFLD is crucial to prevent hepatic damage and other health comorbidities. Currently, the principal treatment for NAFLD is lifestyle modification by diet and exercise, reduction in insulin resistance, and/or treatment of diabetes [5]. The more advanced stages of NASH are thought to need more than lifestyle modification, and several pharmacotherapies are under development. Liver transplantation (LT) is the only treatment option at the stage of decompensated cirrhosis, but with limited access due to increasing organ scarcity. NASH has also emerged as the most rapidly growing cause of hepatocellular carcinoma (HCC) among US patients listed for LT [6], though NAFLD-related HCC diagnosed at more advanced stages (due to less systemic surveillance) can become a contraindication to transplantation.

In recent decades, significant progress has been made to decipher the natural history of fibrosis, fibrogenesis, and microenvironmental conditions that persist in chronic liver diseases. Liver fibrosis arises because of deregulated crosstalk between parenchymal, stellate, and immune cell populations. The initial death of hepatocytes is closely associated with infiltration of immune cells, which increases their secretion of inflammatory molecules, leading to enhanced deposition of collagen and other extracellular matrix proteins in the perisinusoidal space by activated hepatic stellate cells (HSCs). Effective anti-fibrotic therapies aim to prevent the progression of fibrosis to cirrhosis or to induce an improvement in the fibrosis grade. Although several drugs were tested and are currently under development, no effective anti-fibrotic therapy has yet been approved. Advanced and innovative therapies attempt to respond to this unmet medical clinical need [7]. Consistent in vitro and in vivo investigations have revealed the potent paracrine activity of mesenchymal stem cells (MSCs), also referred to as mesenchymal signaling cells, including anti-inflammatory, regenerative, and anti-fibrotic properties [8,9,10]. The beneficial paracrine effects of MSCs are mediated principally by various bioactive molecules, including growth factors and cytokines [9,10]. These multi-target molecules display the capacity of reducing liver inflammation and fibrosis, replenishing functional hepatocytes [11], as well as preventing progressive distortion of the hepatic architecture. MSCs have also been described to modulate several types of immune cells.

Human allogeneic liver-derived progenitor cells (HALPCs) are obtained after primary culture of human parenchymal liver cells isolated from cadaveric donated livers using a classic two-step collagenase perfusion, filtration, and low-speed centrifugation, as previously detailed [12]. HALPCs spontaneously emerge, proliferate, and become predominant at passage 2. HALPCs have been extensively studied at passages 3 to 5 in preclinical studies. Accumulated data have shown that HALPCs display an advanced ability to differentiate into hepatocyte-like cells, as well as potent immunomodulatory and immunosuppressive properties [12,13]. HALPCs have also been successfully manufactured under good manufacturing practices and have been developed as allogeneic products. After being granted medicinal products according to the EU regulation on advanced therapies, their safety was evaluated in three patients with inborn metabolic diseases [14,15] and in a clinical phase I/II trial in pediatric patients with Crigler–Najjar syndrome and urea cycle diseases [16]. Preliminary efficacy was also measured in that trial. Furthermore, the demonstration of their ability to home to the liver after peripheral infusion [17], as well as of their immunomodulatory and anti-fibrotic properties [18,19,20,21], was aligned with the conduction of a phase II clinical study in adult patients with acute-on-chronic liver failure (ACLF) or with acute decompensation at risk of developing ACLF [22]. This trial revealed the ability of HALPCs to ameliorate survival rate, systemic inflammation, and liver functions.

Accordingly, the purpose of the current study is to evaluate the capacity of HALPCs, classified as an advanced therapy medicinal product by the European Medicines Agency, to modulate the immune response and reverse the liver disease state in a mouse model of early and late-stage NASH.

## 2. Materials and Methods

### 2.1. Culture of HALPCs

The human material used to obtain HALPCs is ethically compliant, and its use is rigorously reviewed by highly competent regulatory agencies. The protocol and related experiments were all approved by the competent ethical committees. Written and signed informed consent was obtained for each human liver used. After processing of human cadaveric donated livers thanks to the classical 2-step collagenase (NB1 GMP grade; Nordmark Pharma GmbH, Uetersen, Germany) perfusion, filtration, and low-speed centrifugation, as previously detailed [12], isolated liver parenchymal cells were plated for 24 h under specific culture conditions. Fifteen days later, primary culture medium was substituted with 4.5 g/L glucose Dulbecco’s Modified Eagle’s medium (DMEM) (Invitrogen, Waltham, MA, USA) supplemented with 10% fetal calf serum (FCS) (Gibco, Waltham, MA, USA) leading to HALPCs’ spontaneous emergence. At passage 2, HALPC population becomes predominant and homogeneous, and expands up to passage 5 before cryopreservation. For passaging, HALPCs were enzymatically detached (TrypLE; Invitrogen) and re-plated at 5000 cells/cm^2^. The viability of the recovered cells was always exceeding 95% as determined by trypan blue dye (Sigma-Aldrich, Overijse, Belgium) exclusion method [18,23].

At the end of the expansion step, HALPCs express mainly mesenchymal markers and present low expression of hepatic markers [23]. HALPCs were frozen at passage 5 with CryoStor^®^ CS5 (Sigma-Aldrich) (cryopreservation solution containing 5% dimethylsulfoxide) in liquid nitrogen until use.

### 2.2. Inflammation Priming and Generation of Conditioned Medium

Thawed HALPCs were plated and incubated with DMEM supplemented with 10% FCS in CellBind T flasks (Corning, New York, NY, USA). Twenty-four hours post plating, HALPCs were incubated for 24 additional hours with or without a cocktail of proinflammatory cytokines: 20 ng/mL IL-1β, 10 ng/mL IFN-γ (both from ProSpec Inc., Rehovot, Israel), and 50 ng/mL TNF-α (Peprotech, Rocky Hill, NJ, USA). Thereafter, HALPCs were washed two times with significant volumes of phosphate-buffered solution (PBS) (Invitrogen) to eliminate the added cytokines, and then incubated for 24 additional hours in DMEM supplemented with 0.5% FCS. The conditioned medium (CM) of each experimental condition was recovered, centrifuged to eliminate cell debris, and stored at −80 °C until use.

### 2.3. CD4^+^ T-Cell Activation Assay

Peripheral blood mononuclear cells (PBMCs) were incubated with OKT3 (Fischer Scientific^TM^, Waltham, MA, USA), a monoclonal antibody directed against CD3^+^ lymphocytes, to activate T-cell populations and with CM derived from HALPCs stimulated (or not) with pro-inflammatory cytokines. Cells were plated in 100 µL of X-VIVO medium (with or without OKT3), on top of which 100 µL of CM from HALPC cultures was added (note: we believe that only negligible residual levels of the pro-inflammatory cytokines used for HALPC inflammation priming were left after the dilution of the CM applied here). After 5 days of incubation, the CD4^+^ T-cell population was immunostained and their proliferation analyzed via flow cytometry using CFSE (carboxyfluorescein diacetate succinimidyl ester) dye dilution. The culture medium was also recovered to assess IFN-γ and TNF-α secretion by the immune cells using specific ELISA assays according to the manufacturers’ instructions (IFN-γ—Ref: DY258B and TNF-α—Ref: DY210-05, both from R&D Systems).

### 2.4. Evaluation of Secreted Levels of PGE2, Kynurenine, IFN-γ, and TNF-α in the Supernatants of HALPCs

Prostaglandin E2 (PGE2) and kynurenine measurements: HALPCs were thawed and plated in expansion culture conditions. Twenty-four hours post plating, the cells were incubated for 24 additional hours with or without a cocktail of proinflammatory cytokines (see above) in the presence or absence of COX2 (NS-398) or IDO-1 (INCB024360) inhibitors (both from Abcam). Thereafter, the cell cultures were washed with PBS and incubated in DMEM supplemented with 0.5% FCS for 24 h in the presence or absence of COX2 or IDO inhibitors. The CMs were recovered, centrifuged to eliminate cell debris, and stored at −80 °C until analysis.

The secreted extracellular levels of the studied molecules were measured from the recovered CMs using specific assays according to the manufacturers’ instructions (PGE2—Ref: ab133021 from Abcam, Cambridge, UK; Kynurenine—Ref: K972-100 from Biovision, Waltham, MA, USA, IFN-γ—Ref: DY258B from R&D Systems, Minneapolis, MN, USA; TNF-α—Ref: DY210-05 from R&D Systems).

### 2.5. Transplantation of STAM Mice

C57BL/6 mice were obtained from Japan SLC Inc. (Shizuoka, Japan). All animals used in the study were housed and cared for in accordance with the Japanese Pharmacological Society Guidelines for Animal Use. The experiments conducted on the STAM mice were performed under the ethical committee approval numbers S063 (study number: STMN032-1611-9) and S063-I (study number: SLMN045-1603-4) for early and late-stage NASH, respectively. The animals were housed in TPX cages (CLEA Japan Inc., Tokyo, Japan) with a maximum of 5 mice per cage and maintained in a specific-pathogen-free facility under controlled conditions of temperature, humidity, lighting, and air exchange. NASH was induced in 28 male mice via a single subcutaneous injection of 200 µg streptozotocin (STZ, Sigma-Aldrich, Saint Louis, MO, USA) 2 days after birth and feeding with a solid high-fat diet ad libitum (HFD, 57 kcal% fat, Cat# HFD32, CLEA Japan, Japan) after 4 weeks of age. Indeed, only male STAM mice develop sequential steatohepatitis and fibrosis [24].

For HALPCs (viability ≥ 89%), either single or triple injections of 12.5 × 10^6^ cells/kg body weight (BW) or the vehicle (saline solution 0.9%) were intravenously administered, with or without immunosuppression, at week 6 (1 dose) or weeks 6, 7, and 8 (3 doses) for early-stage NASH, or at week 10 (1 dose) or weeks 10, 11, and 12 (3 doses) for late-stage NASH. Naïve mice were used as controls. Cyclosporine (Sigma-Aldrich), suspended in saline, was administered by intraperitoneal route at a dose of 5 mg/kg every other day. Immunosuppression started at week 5 or week 9 for early and late-stage NASH, respectively.

The mice were observed for significant clinical signs of toxicity, morbidity, and mortality approximately 60 min after each administration. Viability, clinical signs, and behavior were also monitored daily. Body weight was recorded daily during the study period. The animals were sacrificed by exsanguination through direct cardiac puncture under isoflurane anesthesia (Pfizer Inc., New York, NY, USA) at week 9 or week 13 for early and late-stage NASH, respectively.

### 2.6. Blood and Serum Analyses

Blood was collected via cardiac puncture with a 20-gauge needle at necropsy and placed into a Becton Dickinson serum separator tube. For serum analyses, non-fasting blood was collected at euthanasia in serum separate tubes without anti-coagulant and centrifuged at 15,000× *g* for 2 min at 4 °C. The serum was drawn off and aliquoted into Eppendorf cryotubes, then snap-frozen and stored at −80 °C until testing. Serum alanine transaminase (ALT), aspartate transaminase (AST), alkaline phosphatase (ALP), triglycerides, and total cholesterol levels were measured using FUJI DRI-CHEM 7000. Serum CRP was quantified using Mouse High-Sensitive CRP ELISA kit (Kamiya Biomedical Co., Seattle, WA, USA).

### 2.7. Liver Tissue Collection and Processing

At necropsy, after assessing the gross morphology of the liver and overall body condition, the liver was weighed, and the tissue collected for further analyses. For histological analyses, liver samples were fixed in 10% neutral-buffered formalin for 12–18 h (depending on the size of the sample) before being placed into 70% alcohol. Sections were cut from paraffin blocks of liver tissue prefixed in Bouin’s solution and stained with hematoxylin (Muto Pure Chemicals Co., Ltd., Tokyo, Japan) and eosin solution (Wako Pure Chemical Industries, Osaka, Japan). To visualize collagen deposition, Bouin’s fixed liver sections were stained using picro-Sirius red solution (Waldeck GmbH, Münster, Germany). For quantitative analysis of fibrosis area, bright field images of Sirius red-stained sections were captured around the central vein using a digital camera (DFC295; Leica, Wetzlar, Germany) at 200-fold magnification, and the positive areas in 5 fields/section were measured using ImageJ software (National Institute of Health, Bethesda, MA, USA). To visualize macro- and micro-vesicular fat, cryosections were cut from frozen liver tissues, prefixed in 10% neutral buffered formalin, embedded in Tissue-Tek O.C.T. compound (Sakura Finetek Japan Co. Ltd., Tokyo, Japan), and stained with Oil Red O (Sigma-Aldrich).

### 2.8. Statistical Analyses

Results were expressed as mean ± SEM. Statistical analyses were determined by paired Student’s *t*-test for two-sample analyses and one-way ANOVA followed by the Tukey Multiple Comparison Test on GraphPad Prism 9 (GraphPad Software Inc., San Diego, CA, USA), and *p* < 0.05 was considered statistically significant.

## 3. Results

### 3.1. HALPCs Adapt to Inflammatory Environment by Secreting Potent Immunomodulatory Bioactive Molecules

First, the immunomodulatory secretion of HALPCs in normal and inflammatory conditions was assessed in vitro by measuring the levels of two potent bioactive molecules: PGE2 and indoleamine 2,3-dioxygenase (IDO) activity-derived kynurenine, which were shown to be involved in the immunomodulatory activity of MSCs. PGE2 is a lipid, arachidonic acid-derived prostaglandin hormone involved in the regulation of inflammatory responses. IDO is implicated in immune modulation through its ability to limit T-cell function and engage mechanisms of immune tolerance. Our experiments revealed that no impact of inflammation was noticed on the morphological aspects of the cells (Figure 1A). We also reported a decrease in the number of adhering HALPCs following inflammation treatment, as previously documented [23]. The levels of PGE2 (in pg/10^6^ cells) and IDO activity (represented by kynurenine production in nmol/10^6^ cells) were measured in the culture medium. Results revealed that in normal culture conditions, HALPCs secreted basal levels of PGE2 (1723 ± 499 pg/10^6^ cells) (Figure 1B), while the IDO activity was not detectable (0 ± 0 nmol/10^6^ cells of produced kynurenine) (Figure 1C). On the contrary, exposure of HALPCs to pro-inflammatory cytokines for 24 h significantly increased their secretion of PGE2 (126,438 ± 17,132 pg/10^6^ cells) (Figure 1B) and IDO activity (734.9 ± 112.1 nmol/10^6^ cells of produced kynurenine) (Figure 1C).

### 3.2. Effect of HALPC cm on Antigen-Presenting Cell-Induced T-Lymphocyte Proliferation

The proliferation of CD4^+^ T cells was evaluated in the presence of HALPC-derived CM to investigate the potential effect of the immunomodulatory secretion profile of HALPCs. In this assay, OKT3, the soluble anti-CD3 antibody, was used to activate and stimulate the proliferation of CD4^+^ T lymphocytes by the antigen-presenting cells (APCs) (Figure 2A). The PBMCs were cultured with control medium (incubated with no cells) or CM of HALPCs, recovered either in normal non-stimulated condition or after 24 h preconditioning with a cocktail of pro-inflammatory cytokines. Dexamethasone was used as a control for CD4^+^ T-cell inhibition of proliferation. In the CD4^+^ T-cell proliferation assay, the control medium used for the control PBMC culture was a mixture of the medium used for PBMCs and the CM of HALPCs that was not in contact with the cells (either stimulated or not). It corresponded to a mixture of X-VIVO medium (basal medium for PBMC culture) and DMEM supplemented with 0.5% FCS (medium used to generate HALPC CM) at 1:1 ratio (volume). For PBMC treated conditions, HALPC CM (after contact with the cells) was also mixed 1:1 with X-VIVO medium.

As shown in Figure 2B, unstimulated CD4^+^ T cells showed very low levels of proliferation (−anti-CD3 group). Addition of soluble anti-CD3 antibody induced a strong proliferation of CD4^+^ T cells (+anti-CD3 group) which represented the positive reference control (100%). Dexamethasone strongly inhibited the proliferation of CD4^+^ T cells by ~80%. Co-incubation of CD4^+^ T cells with HALPC CM recovered under non-stimulated pro-inflammatory conditions, did not inhibit CD4^+^ T-cell proliferation, whereas co-incubation with HALPC CM recovered after preconditioning for 24 h with pro-inflammatory cytokines significantly reduced the proliferation of CD4^+^ T cells by ~40%. Furthermore, we analyzed the secretion of pro-inflammatory cytokines by activated and proliferating CD4^+^ T cells. As shown in Figure 2C,D, data confirmed the decreased secretion of IFN-γ and TNF-α by more than 50%, respectively, in the CM of HALPCs preconditioned with pro-inflammatory cytokines. Thereafter, we investigated the potential involvement of PGE2 and IDO in inhibiting the activation of APC-induced CD4^+^ T cells by HALPCs. To do so, COX-2 and IDO inhibitors were incubated with HALPCs under normal and pro-inflammatory conditions. The related CM was recovered and incubated with CD4^+^ T cells as mentioned above. As shown in Figure 3A, at both concentrations used, COX-2 inhibitor significantly inhibited the secretion of PGE2 by HALPCs cultured under pro-inflammatory conditions. Inversely, IDO inhibitor at 50 nM specifically and significantly inhibited kynurenine production in HALPCs upon inflammatory culture conditions. As expected, the use of COX-2 inhibitor did not prevent IDO activity (Figure 3B).

We also evaluated the effects of both inhibitors on the proliferation of CD4^+^ T cells. We only measured a minor effect when those inhibitors were used alone or added to the CM of unstimulated HALPCs (Appendix A). When added to the CM of stimulated HALPCs, COX-2 inhibitor significantly restored the proliferation of CD4^+^ T cells compared to the vehicle, which indicates a significant role of the PGE2 pathway. On the contrary, no significant effect of IDO inhibitor was observed on the proliferation of CD4^+^ T cells when added to the CM of preconditioned HALPCs (Figure 3C).

### 3.3. Effect of HALPCs on In Vivo Inflammation and Fibrosis in the NASH-STAM Mouse Model

Based on the immunomodulatory and anti-fibrotic properties of HALPCs previously demonstrated in vitro [19,21], we appraised the potential therapeutic effects of HALPCs in an animal model in which pro-inflammatory signals and fibrosis were involved in the development of the liver pathology. HALPCs were intravenously transplanted, with or without immunosuppression, in the widely used NASH-STAM mouse model at both early and late-stage NASH [25] (Figure 4). The dose and frequency of administration were based on results observed in previous preclinical studies [26] using MSC therapies, and in particular on results of improvement of prognostic and survival scores in clinical trials with patients with chronic inflammatory liver diseases such as NASH [27,28]. Accordingly, repeated doses would be the most appropriate for chronic liver diseases.

The follow-up of mice with early-stage NASH transplanted with HALPCs revealed no significant changes in the mean body weight during the study period and the mean liver weight at euthanasia (week 9) between HALPC-treated mice and vehicle-treated mice in all experimental conditions with or without immunosuppression (Appendix A). Analysis of serum ALT, AST, ALP, triglycerides, and total cholesterol levels between HALPC-treated mice and vehicle-treated mice also revealed no significant changes in all experimental conditions (Appendix A). However, a significant decrease in serum CRP levels was observed in both single- and triple-dose HALPC-treated mice compared to vehicle-treated mice without immunosuppression (Figure 5A). C-reactive protein is an acute phase protein synthesized by the liver, whose circulating concentrations increase in response to inflammation following IL-6 secretion. Persistent high levels of CRP may be observed in chronic inflammatory conditions such as NASH. High-sensitivity CRP (hs-CRP) is considered an obesity-independent surrogate marker of severity of NAFLD, especially the development of NASH [29]. The significant decrease in CRP levels observed in the current study may indicate decreased systemic inflammation related to HALPC administration. Notably, under an immunosuppression regimen, both single- and triple-dose HALPC-treated mice showed a tendency for a dose dependency decrease in CRP levels (although the differences were not statistically significant) (Figure 5B).

NAS is widely used to grade parenchymal alterations and is determined according to the criteria of Kleiner [30]. It is calculated based on the numerical score attributed to steatosis (0–3), hepatocyte ballooning (1–2), and lobular inflammation (0–3), as histologically investigated on fixed liver slices from the analyzed mice. Data showed that liver sections from the vehicle-treated mice exhibited more micro- and macro-vesicular fat deposition, hepatocellular ballooning, and inflammatory cell infiltration compared to normal naïve mice. Although not statistically different, data revealed that transplantation of HALPCs without immunosuppression induced a 19% and 26% decrease in NAS when administered at single (3.80 ± 0.55) and triple doses (3.50 ± 0.47), respectively, compared to the vehicle-treated mice (4.67 ± 0.41) (Figure 5C). With respect to the inflammation score, 2/5 mice treated with one dose of HALPCs showed a ≥50% decrease in the score. In the group of mice treated with three doses of HALPCs, 3/6 reported the same decrease level as the group treated with one dose of HALPCs (Appendix A). Three mice in each group did not show any effect. This high variability in both treated groups is aligned with the absence of statistical significance towards control groups. The same observation was made when hepatocyte ballooning score was analyzed with 2/5 and 4/6 mice that were positively responding after HALPC transplantation (Appendix A).

This decreased tendency seemed to be mostly related to a potential effect on inflammation and hepatocyte ballooning than on steatosis (Appendix A). Under an immunosuppression regimen, only single-dose HALPC-treated mice showed a significant (45%) decrease in NAS (2.20 ± 0.42) compared to related vehicle-treated mice (4.0 ± 0.7). No statistically significant difference was observed when vehicle-treated and triple-dose HALPC-treated mice (3.00 ± 0.28) were compared (Figure 5C). Our data clearly show that with respect to the inflammation score, all analyzed mice treated with one dose of HALPCs had a score of zero, while in the group of mice treated with three doses of HALPCs, only 2/6 mice showed a score of zero (Appendix A). The high variability noticed in the positive control group is aligned with the absence of statistical significance between all analyzed groups. With respect to the hepatocyte ballooning score, 3/5 mice that received one dose of HALPCs showed a 50% decrease compared to the vehicle group, while 4/6 mice that received three doses of HALPCs showed at least a 50% decrease in the score. On steatosis, no clear effect on the cells was measured (Appendix A). Based on those results, we can speculate that, although the data are not statistically significant, the effect of HALPCs was potentially related to a decrease in the hepatocyte ballooning score for both doses, while for the inflammation score, only the one-dose treatment with HALPCs seemed to decrease it.

Sirius red staining was also performed to appraise the fibrosis levels. Among the endpoints investigated in the STAM mouse model to evaluate the usefulness of anti-NASH drugs is the pericentral collagen deposition [31]. Indeed, STAM mice have been reported to show only slight fibrosis after 5 weeks without progression at week 10. Bridging fibrosis as seen in post-carbon tetrachloride (CCl4) treatment can only be observed after 20 weeks in some treated mice [32]. Accordingly, we followed the same protocol previously applied on that mouse model and focused only on that pericentral zone.

Data showed that the vehicle-treated mice displayed higher collagen deposition in the pericentral region of the liver lobule compared to the naïve mice, with or without immunosuppression (Figure 5D). Transplantation of HALPCs without immunosuppression caused a significant decrease (by ~30%) in collagen deposition in the pericentral region when administered at single (0.63 ± 0.08) and triple doses (0.63 ± 0.05), respectively, compared to the vehicle-treated mice (0.90 ± 0.10) (Figure 5D). Under an immunosuppression regimen, collagen deposition was significantly inhibited (−34%) only in mice treated with three doses of HALPC (0.52 ± 0.04) compared to the corresponding vehicle-treated mice (0.80 ± 0.10) (Figure 5D). Analysis of the expression of α-smooth muscle actin in these groups of mice did not reveal any changes related to HALPC transplantation (Appendix A).

Mice with late-stage NASH were transplanted with HALPCs only under an immunosuppression regimen. Post-HALPC transplantation analysis showed no significant changes in the body and liver weight of HALPC-treated mice compared to the vehicle-treated mice (Appendix A). Analysis of serum ALT, ALP, triglycerides, and total cholesterol levels between HALPC-treated mice and vehicle-treated mice also revealed no significant changes in all experimental conditions (Appendix A). Analysis of NAS revealed that HALPC infusion from Week 11 after NASH induction significantly decreased NAS by 25% and 34% when administered at single (3.75 ± 0.39) and triple dose (3.29 ± 0.31), respectively, compared to the vehicle-treated mice (5.0 ± 0.24) (Figure 6A). The improvement in NAS was mainly attributable to a significant reduction in the inflammation score (Appendix A). Sirius red staining and quantification of the positive stained areas revealed that the treatment with HALPCs tended to decrease collagen deposition in the pericentral region by 35% and 40% when administered at single (0.81 ± 0.19) and triple dose (0.75 ± 0.09), respectively, compared to the vehicle-treated mice (1.25 ± 0.21), although the differences were not statistically significant (Figure 6B).

## 4. Discussion

The liver has a tremendous capacity to regenerate, but this potential is hampered by sustained inflammation and consecutive fibrous tissue accumulation. HALPCs are derived from human liver, expanded in vitro, and are currently used in clinical allogeneic development in humans [16,22]. These cells display significant signaling properties thanks to an adaptative secretome and may act as gatekeepers to restore perturbated immune balance under systemic and tissue inflammation. Although HALPCs share similar phenotype characteristics and functional properties with other tissular MSCs, their liver origin is associated with lower immunogenicity [23,33] compared to bone-marrow-derived MSCs for instance, including the lack of constitutive and induced expression of HLA class II molecules, as well as significant expression of HLA g [21] and HLA-E (Unpublished data).

Our first objective was to characterize the secretion of immunomodulatory factors by HALPCs and assess their potential modulating effect on immune cells. We first analyzed the ability of HALPCs to secrete in vitro PGE2 and IDO, two potent bioactive molecules involved in the immunomodulatory activity of MSCs. PGE2 is a lipid, arachidonic acid-derived mediator recognized to display anti-inflammatory [34] and immunomodulatory properties [35] by acting on innate immune cells and by suppressing Th1 differentiation, B-cell functions, T-cell activation, and allergic reaction. Moreover, PGE2 can increase apoptosis of HSCs and inhibit the liver fibrosis in a CCl4 mouse model of fibrosis [36]. IDO is implicated in immune modulation through its ability to limit T-cell function and engage mechanisms of immune tolerance [37]. IDO may also play a protective role against hepatic fibrosis, as was demonstrated in high-fat-diet-induced liver inflammation and fibrosis mouse model [38]. At basal levels, we showed that HALPCs secreted low basal levels of PGE2 and undetectable IDO activity. Inversely, upon proinflammatory conditions, we measured very significant levels of secreted PGE2 and IDO activity in the supernatant of HALPCs. The average level of PGE2 was more than 100,000 pg/10^6^ cells and IDO activity (represented by kynurenine production) was 750 nmol/10^6^ cells). Similar adaptive secretions were observed in previous in vitro studies with other MSCs [39] which highlights the potential immunomodulatory profile of HALPCs.

Thereafter, we evaluated the effect of HALPCs on APC-induced T-cell proliferation. Our data revealed a direct impact of HALPCs on those adaptive immune cells. Indeed, when incubated with the control CM (basal conditions), no significant effect was noticed on CD4^+^ T-cell proliferation. However, when CM of HALPCs previously incubated with pro-inflammatory cytokines was used, a significant reduction in CD4^+^ T-cell proliferation was observed. Such paracrine effects were similarly observed with other MSCs. Williams et al. (2013) [40] showed that human-bone-marrow-derived MSCs do not induce a primary alloantigen-specific T-cell proliferative response, even after IFN-γ-induced upregulation of MHC class II and ICAM-1 on the MSCs, and actively suppress T-cell proliferation induced by either allo-antigens or direct engagement of the T-cell receptor (TCR) and CD28 with activating antibodies. In our study, COX-2 inhibitor significantly restored the proliferation of CD4^+^ T cells when added in the CM of stimulated HALPCs, which stipulates the involvement of the PGE2 pathway. On the contrary, no significant effect of IDO inhibitor was observed on the proliferation of CD4^+^ T cells when added in the CM of preconditioned HALPCs. The dose of IDO inhibitor used in our study has been selected after optimization experiments in which the secretion of kynurenine and of PGE2 were used as primary endpoints. Two different inhibitors were used (INCB024360 and BMS-986205) and 2 different concentrations were tested. Our data clearly demonstrated, as expected, that both inhibitors completely inhibit the secretion of kynurenine at both tested concentrations. With respect to PGE2 secretion, our data revealed that INCB204360 seemed to decrease this parameter in a dose-dependent manner, while BMS-986205 seemed to increase it. Accordingly, we selected the use of INCB204360 at the concentration of 50 nM in our current study. We also studied INCB204360 on the division index of CD4^+^ T cells and demonstrated that the IDO pathway was not involved in the effect observed.

Overall, our in vitro data confirmed that HALPCs are adaptative cells whose activation and secretion can vary under different inflammatory conditions. These observations support the investigations of their therapeutic potential in vivo.

The STAM mouse model is largely used in NASH research and development in which the human NAS system is widely reproduced. Over the past decade, more than 40 papers have been published that used data from the NASH-STAM mouse model to study anti-NASH effects of different compounds as empagliflozin, linagliptin and others [41]. In our study, we assessed the effect of HALPC administration at both early and late stages of NASH development. At early-stage NASH, our data revealed that when no immunosuppression was used, the positive control showed high CRP levels (~4 mg/L) compared to the naïve group. The effect of both single and triple doses of HALPCs significantly decreased CRP levels. With respect to NAS, there was a clear decrease in NAS in three mice and no appreciable effect in two mice (Figure 5C). The same remark applies for the three-dose transplanted group that showed a clear effect of NAS only in three mice. A decrease in fibrosis, as shown in Figure 5D, was clearly noticed for both doses. No effect of HALPCs was noticed on the expression of α-smooth muscle actin, a key marker of activated HSCs. Such result stipulates a direct effect of HALPCs on the extracellular matrix and the promotion of its degradation rather than on HSCs (whose activation seems to be lower than in other models of liver fibrosis). Indeed, MSCs have been reported to adapt their behavior depending on the extracellular environment and to secrete potent metalloproteases such as MMP2 and MMP9, which prevent collagen deposition and accumulation [19,42]. Altogether, these data suggest that the effect of HALPC transplantation in early-stage NASH is potentially both immunomodulatory and anti-fibrotic. Increasing the number of mice per group should confirm these data.

Under an immunosuppression regimen, the positive control did not display high concentrations of CRP as seen in the group without immunosuppression (Figure 5B). Furthermore, high variability was seen in that group, which could explain the absence of statistical significance, although the trend could be observed at least for the three-dose transplanted group. The same observation applies for NAS under an immunosuppression regimen (Figure 5C). The positive control showed high variability, which impacted the statistical significance for the three-dose transplanted group (although the decreasing trend is quite clear in 5/6 mice). For fibrosis analysis, the effect of three doses of HALPCs is quite significant and could be acceptably aligned with the trended decrease in CRP levels and NAS.

In early-stage NASH, the decreased tendency was mostly related to a potential effect on inflammation and hepatocyte ballooning (Appendix A). However, no effect on steatosis was measured.

For late-stage NASH, the study was conducted with immunosuppression to increase the potential survival of transplanted cells. The effect of HALPCs on NAS was clearly significant for both tested doses. For fibrosis analysis, quantification of Sirius red staining did not show statistical significance, although for the single and triple dose HALPC-treated mice, 4/8 and 5/7 transplanted mice showed a clear decrease, respectively (mean decrease of ~30% in the pericentral region). We believe that the high variability seen in the positive control significantly impacted this parameter. Nevertheless, for both early and late-stage NASH, there was always a positive correlation between all immune and fibrosis parameters, although this was not statistically significant in some cases (Appendix A). These findings are important, because a reduction in NAS is one of the major clinical endpoints for assessing the drug efficacy in NASH patients [43]. Thus, this result suggests a potential effect of HALPCs as anti-NASH therapeutics.

Previous experiments with similar cells produced in an academic setting demonstrated their direct inhibitory effect on activated HSCs and on the expression of markers of liver fibrosis in rats [19]. In NASH, the fibrosis stage is critical, as it is the most important predictor of liver-related mortality. Accordingly, preventing fibrosis in NASH patients will prevent both cirrhosis and HCC. A large study with more than 30 years of follow-up in NAFLD patients demonstrated that overall mortality was not increased in patients with NAS 5–8 and fibrosis stage 0–2 (HR 1.41, CI 0.97–2.06, P50.07), whereas patients with fibrosis stage 3–4, irrespective of NAS, had increased mortality (HR 3.3, CI 2.27–4.76, *p* < 0.001) [44].

One of the limitations found in these in vivo models was the use of immunosuppressors to avoid a potential rejection of the xeno-transplanted cells. Importantly, our study clearly revealed that the immunomodulatory and anti-fibrotic effects of HALPCs demonstrated in the STAM mouse model were observed with and without immunosuppression. There was no difference between vehicle-treated mice with or without immunosuppression at early stages of the disease. Immunosuppression did not by itself affect the disease progression in this model. For this reason, immunosuppression has been used in the advanced stage of this NASH model.

Quantification of immune cell populations, including T and NK cell populations, serum antibodies, and pro- and anti-inflammatory cytokines would be of great interest for a deep understanding of the modulation of the immune system after allogeneic transplantation in the STAM mouse model. Considering that intra-hepatic Kupffer cells recruit neutrophils, monocytes, and NK cells [45], it would be of interest to characterize this innate system, and thereafter assess the impact of HALPC transplantation.

Understanding the complex pathways in the pathophysiology of NASH is a challenge, and one of the challenges of pharmacotherapy is the increasing complexity of the disease as it progresses.

Since there is still an unmet medical need for patients not achieving NASH resolution, several molecules targeting different pathways have been developed in the past few decades, with the aim of stopping the progression of liver diseases [46]. However, most of the current developed anti-NASH therapeutics act, mechanistically, at a single level in the pathogenesis of the disease. A few of those tested small molecules can simultaneously act at different levels. In our study, we demonstrated that the intravenous administration of HALPCs exerts significant both anti-inflammatory and anti-fibrotic effects in the NASH-STAM mouse model, which is aligned with the in vitro documented properties of HALPCs [19,47]. These features suggest potential multi-target action of HALPCs in complex pathogenesis, which may modify the progression course of fibro-inflammatory liver diseases.

## Figures and Tables

**Figure 1 cells-11-02854-f001:**
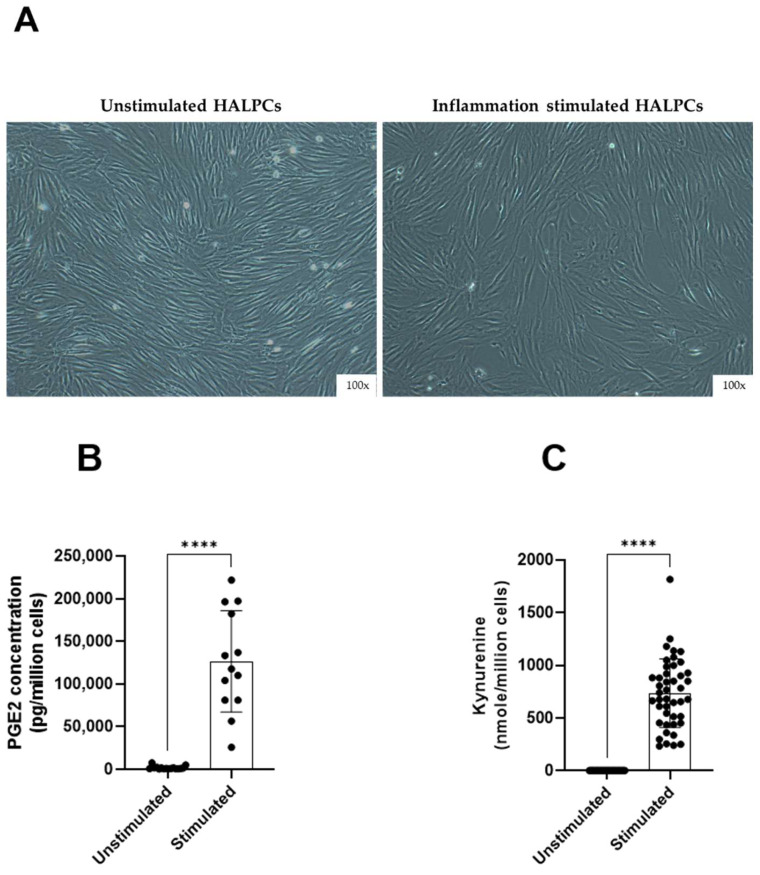
Secreted levels of immunomodulatory factors by HALPCs. (**A**) Morphology of HALPCs (representative sample) observed microscopically after treatment with the cocktail of proinflammatory cytokines (*n* ≥ 3 samples from different donors). Magnification: 100×. (**B**) Prostaglandin E2 (PGE2) secreted levels. (**C**) IDO activity represented by kynurenine secretion in the CM of HALPCs). Unstimulated: absence of pro-inflammatory cytokines (*n* = 16); stimulated: presence of pro-inflammatory cytokines (*n* ≥ 13). ****: *p* < 0.0001 (Unpaired *t*-test).

**Figure 2 cells-11-02854-f002:**
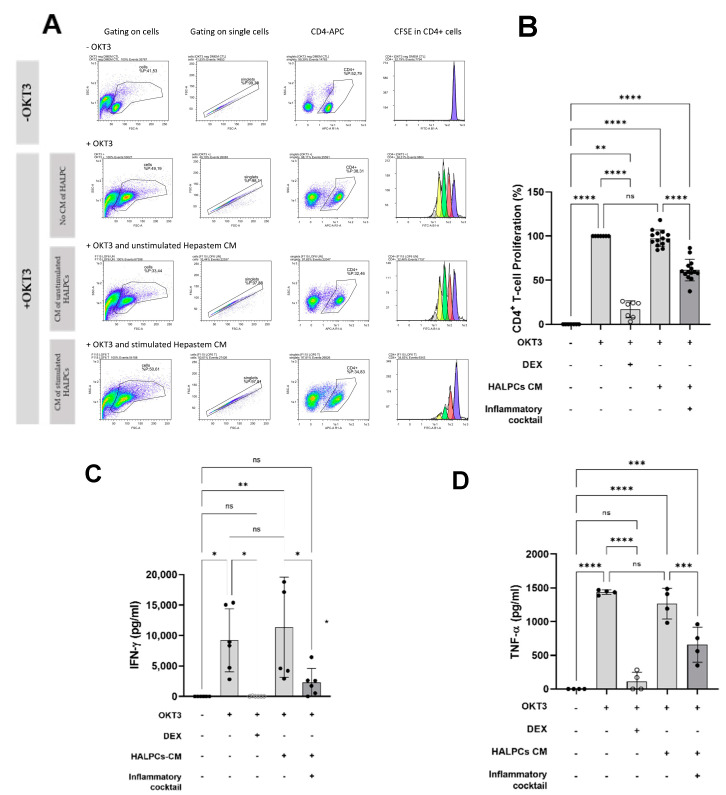
Effect of HALPCs conditioned medium on APC-induced T-lymphocyte proliferation. (**A**), Representative experiment showing the plot for the different control and tested conditions: Scatter plot FSC/SSC, singlets, CD4^+^ T cells, CFSE staining. The number of peaks (progressive dilution) of CFSE linked to CD4^+^ T-cell proliferation are clearly reduced in the treated compared to the control condition (*n* ≥ 3). (**B**), Contact-independent HALPC inhibition of APC-induced CD4^+^ T-cell proliferation in pro-inflammatory conditions (*n* = 7). (**C**), Effect of CM of HALPCs on IFN-γ secretion in APC-induced CD4^+^ T-cell activation assay (*n* = 6). (**D**), Effect of CM of HALPCs on TNF-α secretion in APC-induced CD4^+^ T-cell activation assay (*n* = 4). HALPCs: human allogeneic liver-derived progenitor cells; CM: conditioned medium. ns: not significant; *: *p* < 0.05; **: *p* < 0.01; ***: *p* < 0.001; ****: *p* < 0.0001 (Tukey Multiple Comparison Test).

**Figure 3 cells-11-02854-f003:**
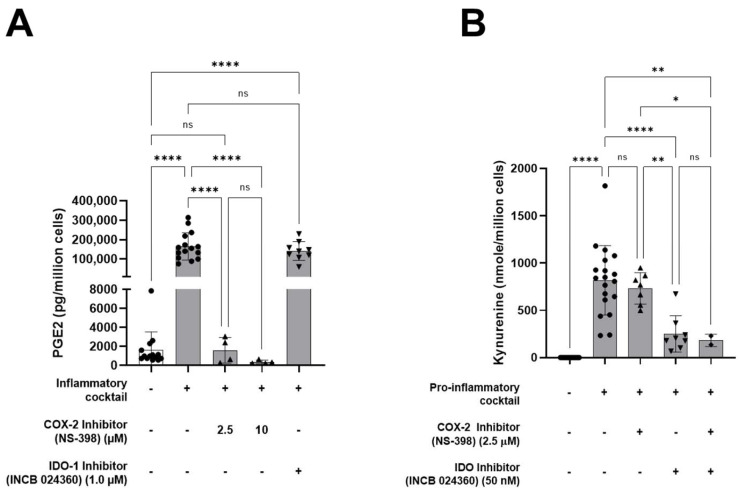
Mechanism(s) of action involved in the inhibition of APC-induced CD4^+^ T-cell activation by HALPCs in pro-inflammatory conditions. (**A**) Effect of IDO and COX-2 inhibitors on PGE2 secretion by HALPCs (*n* ≥ 3). (**B**) Effect of IDO and COX-2 inhibitors on kynurenine production by HALPCs (*n* ≥ 3). (**C**) Effect of IDO and COX-2 inhibitors on inhibition of APC-induced CD4^+^ T-cell proliferation by HALPCs (*n* ≥ 3). HALPCs: human allogeneic liver-derived progenitor cells; CM: conditioned medium. ns: not significant; *: *p* < 0.05; **: *p* < 0.01; ***: *p* < 0.001; ****: *p* < 0.0001 (Tukey Multiple Comparison Test).

**Figure 4 cells-11-02854-f004:**
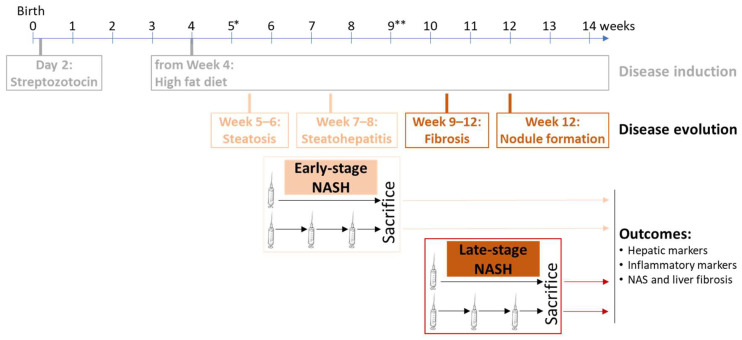
STAM mouse model and HALPC transplantation protocol applied. NASH was induced in C57BL/6 male mice by a single subcutaneous injection of 200 µg streptozotocin 2 days after birth and by solid high-fat diet ad libitum after 4 weeks of age. HALPCs (either single or triple injections of 12.5 × 10^6^ cells/kg body weight) or the vehicle (saline solution 0.9%) were administered via the tail vein at week 6 (1 dose) or weeks 6, 7, and 8 (3 doses) for early stage NASH, or at week 10 (1 dose) or weeks 10, 11, and 12 (3 doses) for late-stage NASH. Mice viability, clinical signs, and behavior were monitored daily. The mice were sacrificed by exsanguination through direct cardiac puncture under isoflurane anesthesia at weeks 9 and 13 for early and late-stage NASH, respectively. Immunosuppression (with cyclosporine 5 mg/kg every other day) started at *Week 5 for early-stage NASH or at **Week 9 for late-stage NASH.

**Figure 5 cells-11-02854-f005:**
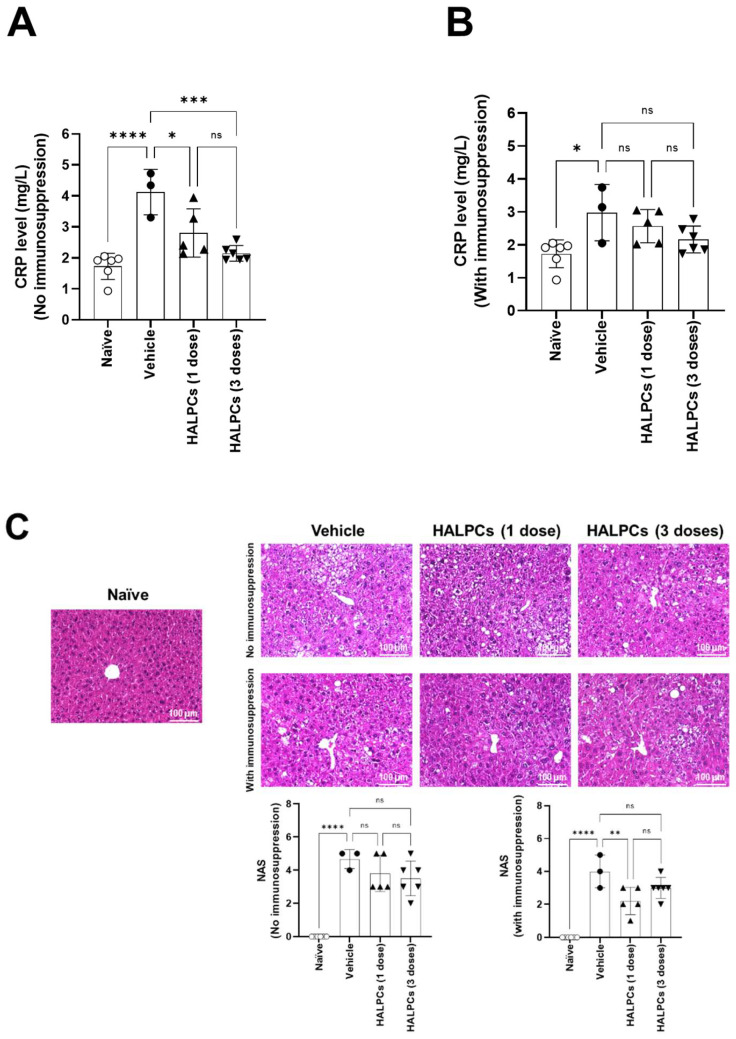
Evaluation of HALPC transplantation efficacy in early-stage NASH mouse model. Serum CRP was quantified by Mouse High-Sensitive CRP ELISA kit on non-fasting serum from the different mice groups. (**A**) Without immunosuppression regimen. (**B**) With immunosuppression regimen. (**C**) Representative photomicrographs of HE-stained liver sections. NAFLD Activity score (NAS) was calculated according to the criteria of Kleiner [30]. (**D**) Representative photomicrographs of Sirius red-stained sections. Bright field images of Sirius red-stained sections were captured around the central vein using a digital camera at 200-fold magnification, and the positive areas in 5 fields/sections were measured using ImageJ software (National Institute of Health, USA). ns: not significant; *: *p* < 0.05; **: *p* < 0.01; ***: *p* < 0.001; ****: *p* < 0.0001 (Tukey Multiple Comparison Test).

**Figure 6 cells-11-02854-f006:**
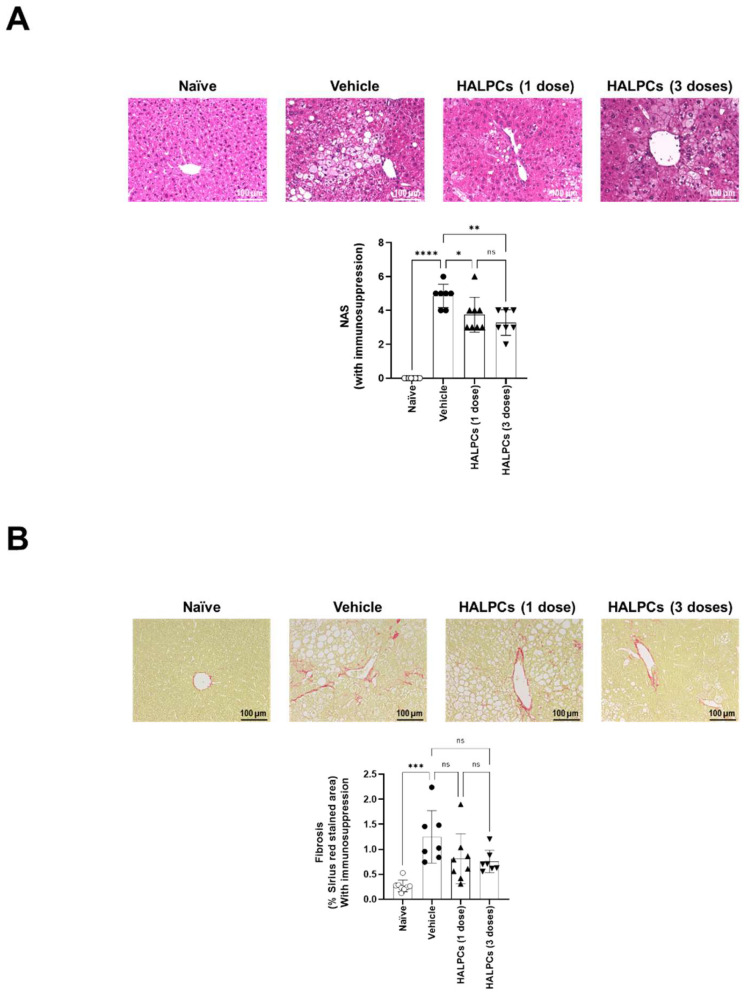
Evaluation of HALPC transplantation efficacy in late-stage NASH mouse model under immunosuppression regimen. (**A**) Representative photomicrographs of HE-stained liver sections. NAFLD Activity Score (NAS) was calculated according to the criteria of Kleiner [30]. (**B**) Representative photomicrograph of Sirius red-stained sections. Bright field images were captured around the central vein using a digital camera (DFC295; Leica, Germany) at 200-fold magnification, and the positive areas in 5 fields/section were measured using ImageJ Software (National Institute of Health, USA). ns: not significant; *: *p* < 0.05; **: *p* < 0.01; ***: *p* < 0.001; ****: *p* < 0.0001 (Tukey Multiple Comparison Test).

## Data Availability

Data are available upon reasonable request.

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
