# Peer review of "Human Allogeneic Liver-Derived Progenitor Cells Significantly Improve NAFLD Activity Score and Fibrosis in Late-Stage NASH Animal Model"

_cells, 2022, doi:10.3390/cells11182854_

Round 1

Reviewer 1 Report

NASH is one of the most concerning diseases in the last few years, and with the close association with other disorders such as obesity and diabetes, this concern is increasing exponentially. Additionaly, human allogeneic liver-derived progenitor cells (HALPCs) have gained many supporters due to the large amount of clinical evidence supporting their use for the treatment of fibro-inflammatory liver diseases. The aim of the present study was to analyze the anti-inflammatory and anti-fibrotic characteristics of HALPCs in a mouse model of NASH for potential reversal of this chronic liver disease.

In my opinion, topic of manuscript is suitable for Cells. I really appreciate that the authors have shown that prostaglandin E2 is directly involved in the regulation of CD4+T cell proliferation and their secretion of proinflammatory cytokines. In addition, the authors demonstrated that transplantation of HALPCs improved inflammation and fibrosis through decreased collagen deposition in early and, especially, late-stages of NASH. Nevertheless, the following points need to be answered:

1.     I strongly recommend that the authors include the results obtained using the COX-2 and IDO inhibitors alone and adding the CM of unstimulated HALPCs in Figure 3. I think they are a good negative control results.

2.     Regarding Figure 3, the authors mentioned that “no significant effect of IDO inhibitor was observed on the proliferation of CD4+ T cells when added in the CM of preconditioned HALPCs”. However, there was a tendency to recover some CD4+ T cells proliferation after addition of the IDO inhibitor. Perhaps higher concentrations are needed. Why did the authors choose these doses? It is true that the high dose of COX-2 inhibitor (10uM) shows a higher reduction of PGE2 concentration in comparison with the reduction of kynurenine with the dose of IDO inhibitor used (50nm). The authors should test higher concentrations of the IDO inhibitor in order to obtain a greater reduction of kynurenine. In this way, they could analyze better the possible effect of the IDO inhibitor in the CD4+ T cells proliferation.

3.     The authors should include an explanation in the main text about why they selected to observe the effect of the HALPCs transplantation in early and late-stage NASH but not in hepatocellular carcinoma (HCC). Especially, when they included the HCC period in Figure 4.  

4.     Regarding Figure 5, this figure is not clearly visible. How do the authors explain the different trend in the effect of HALPCs transplantation in early-stage NASH analyzing different features such as fibrosis, CRP values and NAS score?

5.     As for late-stage NASH, again figure 6 is not clearly visible. In addition, the authors mentioned in the text that: "in the early-stage NASH model, follow-up of transplanted mice revealed no significant changes in mean body weight during the study period and mean liver weight at euthanasia.... Analysis of serum levels of ALT, AST, ALP, triglycerides and total cholesterol between HALPC-treated and vehicle-treated mice also revealed no changes". But what results were obtained in the late-stage NASH model? This is not mentioned in the text and could give us more information about the therapeutic effect of HALPC transplantation, as well as its possible toxic effects.

6.     I strongly believe that the authors should include an experiment analyzing immune responses against allogeneic cell transplantation with or without immunosuppression. Perhaps, the authors should quantify T and NK cell populations, serum antibodies, analyse pro and pre-inflammatory cytokines in serum by ELISA. In addition, it would be interesting to analyse the effect of HALPCs transplantation on hepatic steatosis.

Additionally, the following points require to be taken into consideration:

1.     Authors should include some references in the introduction (lines 72-73).

2.     Authors should be careful with the extra spaces and italics in some words such as in vitro and in vivo.

3.     What type of collagenase did the authors use? (line 93).

4.     Why did the authors select single or triple injections and not double injections? Why did they choose 12.5x106 cells/Kg (line 135)?

5.     Why did the authors choose male mice instead of female mice (line 131)?

6.     The authors mentioned: “the vehicle-treated group displayed higher collagen deposition in the pericentral region of liver lobule..” (line 293). What about the midlobular or centrilobular region of the hepatic lobe?

7.     Why did the authors not include the “no immunosuppression group” in the late-stage NASH mouse model?

Reviewer 2 Report

1. The abstract should be less than 200 words. Please cut length accordingly.

2. The background of HALPCs was not well addressed. Please enrich the Introduction with the definition, characterization and existing applications of HALPCs. 

3. Any specific reason why the author used CS5 instead of CS10 for the cell freezing? Just curious.

4. The culture method of HALPCs, if not developed originally, should be supplemented with background and references.

5. Figure 1 should be supplemented with bright-field/phase contrast images of HALPCs after different treatments.

6. Figure 2 is hard to read. Please use images with higher resolution.

7. What medium was the control cells cultured with in the HALPCs-CM treatment test (Figure 2)? If the control cells were not cultured with the same medium that was used to make the HALPCs-CM, the comparison here would be questionable.

8. A flow profiling of the T cells after treatment should be included in Figure 2.

9. Scale bars should be added in all staining images. And images with higher resolution should be used to replace the existing ones. 

10. How was the HALPC transplantation dose defined? Was there a comparison between different doses?

Reviewer 3 Report

In this study the authors investigated the antifibrotic effect of the EV administration of human allogeneic liver-derived progenitor cells (HALPCs) in  NASH mice with different stages of liver disfunction.

The study is very interesting, but only preliminary investigations have been done, thus further experiments need to be performed to support the antifibrotic effect and the proposed mechanism observed in in vitro investigations.

Maior concerns:

1) the abstract needs to be summarized to fit to 200 words and it's better to divide it using the subtitles, e.g. background, methods...

2) lines 93-94, even though  the used method to extract HALPCs is published elsewhere, add some detail on the provenience of human samples.

3) in the section "CD4+ activation assay", the authors have to detail how much CM was used, if it was mixed with fresh medium and the concentration of cytokines used to induce inflammation of HALPCs and how they measured IFN and TNF in cell supernatant, e.g. detailing the kit used for this assay. 

4) line 136. the authors have to improve and specify how they have induce immunosuppression in mice.

5) line 250. the authors stated that they have demonstrated antifibrotic activity of HALPCs in vitro, nevertheless they didn't demonstrated it using as example activated hepatic stellate cells. They have reported a proof of anti-inflammatory in vitro activity.

6) In Fig. 2 they reported the quantification of IFNg and TNFa. These values seems to be different if normalized to the increased expansion of CD8+ T cells treated with inflamed CM of HALPCs. Have the authors have an explanation or an hypothesis to improve this part of the discussion? Moreover, they have taken into account that in with the inflammatory cocktail they added these two cytokines to HALPC medium thus changes basal content of IFNg and TNFa in the medium? Have they check it? 

7) The authors have do an immunosuppressive treatment in NASH mice only for early stage. They have to detail the reason of immunosuppression only in this early stage model in the manuscript.

8) They authors measure body weight and some plasma indexes stating that they haven't observed any difference. Otherwise this reviewer think that these data need to be added in the manuscript, e.g. as supplementary, to give more insight of the model used in this study, and also to give the variation observed between healthy and NASH mice.

9) lines 288-291. the author wrote "The decreased tendency seems to be mostly related to a potential effect on inflammation and hepatocyte ballooning than on steatosis (Supplementary Figure 1B)". nevertheless if there is no statistically significant difference they couldn't demonstrate this hypothesis.

10) the authors have performed only histological evaluations on hepatic tissue of NAS animals. They have to add an analysis on activation of HSCs, e.g. IHC on aSMA.

11)The authors have also to evaluate some circulating interleukins to further confirm their hypothesis on the mechanism of action of HALPCs in vivo, and also a flow cytometry analysis of CD8+ T cells and check their activation in vivo.

12) The authors have to improve the discussion on the possible explanation of the different behavior of one or triple administration of HALPCs in NASH animals.

Minor concerns:

1) please check the English, since many errors along the manuscript have been detected, e.g. lines 210, 251 and so on.

2) the references need to be check since some refuses are present in the text, e.g. line 503.

3) line 86 please add the acronym for ACLF

4) lines 118, please add the detail of supplier of materials used in the assays and add the number of animal study authorization.

5) refuse in line 426.

Round 2

Reviewer 2 Report

The manuscript does not contain any figures.

Author Response

Dear Reviewer,

we apologize for this. The revised version of the figures and supplementary material have been uploaded separately

We do now insert them after the references list in the clean word version and also send them to you in a separate pdf file for your kind perusal

Best regards

Reviewer 3 Report

The authors have improved the manuscripts, nevetheless the figures were removed from the revised version and the supplementary files are partially not uploaded and lacking, thus I could not judge all the modifications the authors have done to the manuscript. 

As regarding the reply to point 6) they have to stated in the MM they have performed a wash to eliminate added cytokines.

As regarding the reply to point 8), the authors haven't included in the supplementary body weight and plasma indexes as stated in the reply. 

As regarding point 9), please highlight the change in the text that seems not to be changed.

As regarding point 10), I haven't found the supplementary table 1. 

Some typing errors are still present, e.g. square brackets instead of round ones in the added references.

Author Response

Dear Reviewer,

we apologize for this. The revised version of the figures and supplementary material have been uploaded separately

We do now insert them after the references list in the word version and also send them to you in a separate pdf file for your kind perusal

We will address the new concerns separately after finalizing your review

Thank you again 

Round 3

Reviewer 2 Report

Accept in present form

Author Response

Dear reviewer,

Thank you for your constructive comments that significantly improve our manuscript

Respectfully yours,

Reviewer 3 Report

We thank the authors for the new additional files.

Some concerns need to be addressed before acceptance.

As regarding the reply to point 6) of the first revision, they have to stated in the MM they have performed a wash to eliminate added cytokines.

As regarding point 9) of the first revision, please highlight the change in the text that seems not to be changed. Please reformulate adding some reason the sentence "The decreased tendency seems to be mostly related to a potential effect on inflammation and hepatocyte ballooning than on steatosis (Supplementary Figure 1B). "

As regarding the fact the authors haven't observed any change in aSMA gene expression, they have an explanation? How do they correlate this with the other effect on fibrosis Please add some explanation in the discussion. 

Some typing errors are still present, e.g. square brackets instead of round ones in the added references.

Author Response

Dear Reviewer,

We sincerely thank you for your constructive comments that significantly help us to improve our manuscript.

The answers to your concerns are included in the attached file

Respectfully yours,  
